# The Influence of ZnO Oxide Layer on the Physicochemical Behavior of Ti6Al4V Titanium Alloy

**DOI:** 10.3390/ma14010230

**Published:** 2021-01-05

**Authors:** Anna Woźniak, Witold Walke, Agata Jakóbik-Kolon, Bogusław Ziębowicz, Zbigniew Brytan, Marcin Adamiak

**Affiliations:** 1Department of Materials Engineering and Biomaterials, Faculty of Mechanical Engineering, Silesian University of Technology, Konarskiego 18A Street, 44-100 Gliwice, Poland; boguslaw.ziebowicz@polsl.pl (B.Z.); zbigniew.brytan@polsl.pl (Z.B.); marcin.adamiak@polsl.pl (M.A.); 2Department of Biomaterials and Medical Devices Engineering, Faculty of Biomedical Engineering, Silesian University of Technology, Roosevelta 40 Street, 41-800 Zabrze, Poland; witold.walke@polsl.pl; 3Department of Inorganic, Analytical Chemistry and Electrochemistry, Faculty of Chemistry, Silesian University of Technology, B. Krzywoustego 6 Street, 44-100 Gliwice, Poland; agata.jakobik-kolon@polsl.pl

**Keywords:** Ti6Al4V titanium alloy, corrosion test, EIS test, contact angle, inflammatory, ALD (atomic layer deposition), zinc oxide ZnO

## Abstract

Titanium and its alloys are characterized by high biocompatibility and good corrosion resistance as a result of the ability to form a TiO_2_ oxide layer. However, based on literature data it can be concluded that titanium degradation products, in the form of titanium particles, metal-protein groups, oxides and ions, may cause allergic, inflammatory reactions and bone resorption. The corrosion process of Ti6Al4V in the human body environment may be intensified by a decreased pH and concentration of chloride compounds. The purpose of this article was to analyze the corrosion resistance of the Ti6Al4V alloy, obtained by the selective laser melting method in a corrosion solution of neutral pH and in a solution simulating peri-implant inflammatory conditions. Additionally, the influence of zinc oxide deposited by the atomic layer deposition method on the improvement of the physicochemical behavior of the Ti6Al4V alloy was analyzed. In order to characterize the ZnO layer, tests of chemical and phase composition as well as surface morphology investigation were performed. As part of the assessment of the physicochemical properties of the uncoated samples and those with the ZnO layer, tests of wetting angle, pitting corrosion and impedance corrosion were carried out. The number of ions released after the potentiodynamic test were measured using the inductively coupled plasma atomic emission spectrometry (ICP–AES) method. It can be concluded that samples after surface modification (with the ZnO layer) were characterized by favorable physicochemical properties and had higher corrosion resistance.

## 1. Introduction

Generally, titanium and its alloys haves been regarded as inert and biocompatible materials, which are ideal for long-lasting implantation in medical applications [1,2,3,4]. Good corrosion resistance is associated with the ability to spontaneously form a thin and stable protective oxide layer, which constitute a compact and dense kinetic barrier for extensive corrosion [5,6,7]. However, mechanical stress results in depassivation, bare metal exposure, repassivation and corrosion [8,9,10]. In vivo corrosion and tribological wear of titanium materials can, therefore, lead to the release of particles in the form of TiO_2_, inorganic metallic salts, as well as free metal ions to the intraarticular joint space. Unfortunately, and contrary to popular belief, released titanium degradation products may cause serious side effects such as inflammation, pain [11], cytotoxicity [12,13], allergy [14], genotoxicity [15], carcinogenicity [16]. Additionally, titanium dioxide TiO_2_ can alter the viability and behavior of multiple bone cells, which may result in bone resorption, aseptic implant loosening and disrupt implant retention [17,18]. On this basis, it can be concluded that the chemical and physical properties of the surface of the medical device are responsible for the proper integration of the implant with the environment of the human body. In recent years, the use of inorganic oxide layers in biomedical application has attracted interest due to their high chemical stability, satisfactory biocompatibility and possible high antibacterial potential. The most popular metal oxide layers are: silver Ag_2_O [19], copper oxide CuO [20], gold Au_2_O_3_ [21], titanium oxide TiO_2_ [22]_,_ aluminum oxide Al_2_O_3_ [23]_,_ zirconia oxide ZrO_2_ [24]_._ Currently, there are some reports on the antibacterial activity and corrosion protection of ZnO [25]. Mao et al. [26] pointed to the effect of zinc ions on enhancing antibacterial activities and stimulating the immune function of the human body. Based on the results obtained by Foroutan [27], it can be concluded that the ZnO nanoparticles stimulated the process of differentiation of MSCs (mensenchymal stem cells), which depended on the dose as well as on the size of ZnO. According to the studies performed by Bowen et al. [28], it can be concluded that the Zn^2+^ ions released from a zinc implant may suppress restenosis pathways and exhibit excellent biocompatibility with the arterial tissue, which was also confirmed by Yang et al. [29].

In many literature reports the effect of surface conditions on the physicochemical properties, especially corrosion resistance of Ti-based alloys, has been investigated, but in many cases the corrosion tests were performed in a neutral solution of pH = 7–8 [30,31]. However, interference with the human body environment disturbs its existing state of balance and thus triggers defense reactions, such as decreased pH of body solution and increased temperature. In some reports, in order to simulate relevant clinical scenarios, the pH of the solution was decreased by the addition of fluoride ions [32,33]. However, there is no clear information about the influence of chloride ions on the corrosive behavior of titanium alloys. Oshida et al. [34] suggested that implantological success depends to a large extent on pH values and the concentration of oxygen or chloride compounds. The corrosion of the metal implant can also be accelerated by the adhesion of bacteria [35]. Therefore, the authors of this work proposed a modification of the Ti6Al4V alloy surface by deposition of a ZnO layer by the atomic layer deposition (ALD) method. The ALD method allows deposition of extremely conformal and high-quality barrier layers with controllable thickness, even on complex three-dimensional surfaces. The bone implants are often characterized by very complex shapes, in many cases obtained by selective laser melting (SLM) methods, which does not appear to be a limitation from the designer’s point view [36,37]. The aim of this study is to understand the mechanism underpinning the effect of Ringer’s solution pH on the corrosion of the Ti6Al4V alloy and eventually the protective action of the ZnO oxide layer. The quality of the ZnO layer was investigated by microscopic observations, using scanning electron microscopy (SEM) and atomic force microscopy (AFM) as well as phase investigation by X-Ray Diffraction method. In order to reflect possible changes in degradation of the uncoated metal surface and the surface with ZnO layer, the potentiodynamic tests were performed. Additionally, to test processes and phenomena occurring at the interface between the implant and the tissue environment (in order to characterize the metal/oxide/solution interface), electrochemical impedance spectroscopy (EIS) tests were performed. The results of the research were supplemented with the analysis of the amount of ionized ions released to the corrosive solution using inductively coupled plasma atomic emission spectrometry (ICP–AEM) method. Wetting angle measurements and surface free energy (SFE) calculations were performed to obtain the information about the chemical character of the surface, which are important factors that could affect cellular response and bacterial plaque accumulation. The selection of the investigated aspects is related to the functioning of the implants.

## 2. Materials and Methods

The subject of the study was the Ti6Al4V titanium alloy, manufactured by SLM. The chemical composition delivered and used in the experiments met the requirements of ISO 5832-3 [38] and ASTM F-136-13 [39] standards and is shown in Table 1.

It was found that the chemical composition of the Ti6Al4V powder and as-fabricated samples were almost the same as the declared requirements. A similar elements content within the powder and as-fabricated specimens designated that the alloying elements evaporation during laser interaction with powder material during the SLM process was negligible. The Ti6Al4V powder was spherical in shape (Figure 1a).

The powder particle size was in the range of 19–65 µm (Figure 1b), as measured using a laser particle size analyzer Analysette 22 MicroTec PLUS (Fritsch, Sonneberg, German). The median diameter D50 was 35.86 µm, and D10–D90 diameter range was 19.12–60.75 µm. The shape of the particle size distribution scheme was relatively narrow and symmetric. The Ti6Al4V cube with dimensions of 10 × 10 × 10 mm was built using a laser-based SLM machine AM250 (Reinshaw, New Mills, UK), which is employs a Ytterbium fiber laser (Yb) with a wavelength of λ = 1074 nm (laser power P—400 W, scanning speed SP—up to 2000 mm/s). The tested samples were manufactured using the laser power of 400 W and laser speed of 500 mm/s within the protective atmosphere of high purity argon (99.996%). The tested samples were manufactured using a meander scanning strategy, and the direction of scanning was rotated through 67° between successive layers. In order to guarantee good adhesion of the powder material to the base platform, before the start of the process the plate was subjected to polishing. Process parameters were designed with MARCAM AutoFab software (PresseBox, Baden-Württemberg, Germany). Final tested samples were subjected to heat treatment at T = 800 °C for t = 6 h.

After processing, the tested samples were subjected to mechanical finishing, using the grinding-polishing machine TERGAMIN-30 (Struers, Willich, Germany). The samples were ground with SiC abrasive papers of 4000 and 2400 grain/mm^2^ and polished, which was performed using a colloidal silica suspension OP-U 0.04 µm. Next, the samples were thoroughly cleaned in acetone and then dried. After mechanical finishing the samples were in the initial state (M1) for the tests. The zinc oxide layer ZnO has been deposited by an ALD (Atomic Layer Deposition) method, using an advanced ALD system R-200 (Picosun, Masalantie, Finland). Diethylzinc (DEZ) was used as a precursor for zinc, and deionized water as a precursor for the oxide. Nitrogen was used as an inert gas. The dosing time for each of the precursors was 0.1 s. The flushing time was 4 s for DEZ and 5 s for deionized water. During the ALD process, the chemical reaction was thermally assisted and the temperature was set at 150 °C, which is located in the so-called “growth window”. The number of cycles was 1500. The thickness of the layer was approximately ~200 ± 20 nm. The samples after the ALD process were the second group M2 for the tests.

The surface observation of the tested samples was performed using the scanning electron microscope Supra 35 (Zeiss, Oberkochen, Germany), equipped with type InLess detector. The microscopic observations were conducted at 70,000 magnification with an accelerating voltage of 5 kV.

The surface morphology and topography were examined using AFM XE—100 atomic force microscopy (Park System, Suwon, Korea). The observation was conducted by the contact mode, and the scan area was 10 × 10 µm^2^.

The phase constituents of samples with the ZnO layer were examined by the X-ray diffraction method, using X’Pert PRO X-ray diffractometer (Panalytical, Almelo, The Netherlands) using Co Kα radiation with 2θ ranging from 20° to 90° with a step of 0.05. The lamp was set to 40 kV and the heater current of 30 mA was used. To identify the ZnO phase components, a thin layer of zinc oxide was applied to a SiC substrate to remove interference from the substrate material.

The EIS test consists in determining the response of the tested electrical system at a determined free potential and current pulse [40]. An electrochemical examination was performed using the test stand comprised of an AutoLab’s PGSTAT 302N system (AutoLab, Warsaw, Poland), equipped with the frequency response analyzer (FRA2) module and an electrochemical cell with a three-electrode system, where the tested sample was used as a working electrode (anode), an auxiliary electrode was represented by a platinum wire (PtP-201), and the reference electrode was a saturated Ag/AgCl electrode. The tests were performed within a 10^4^–10^−3^ Hz frequency range, and the voltage amplitude of the sinusoidal signal activating amounted to 10 mV. In the test, impedance spectra of the system were determined and the measurement data were compared against the equivalent system. The impedance spectra of the studies are presented in the form of Nyquist diagrams for different frequency values and Bode diagrams. The obtained measurement data were adjusted through the method of the smallest squares to the substitute setup and the values of resistance R and capacity C were determined.

The potentiodynamic test by recording anodic polarization curves, according to PN-EN ISO 10993-15 [41] standard was performed. The measurements were taken with the use of Atlas 0531 EU potentiostat (ATLAS-SOLLICH, Rębiechowo Poland) with a PC with the AtlasLab software and three electrode system, identical to that used in the EIS test. At first, the open circuit potential (OCP) was established at electrolysis conditions, after which the potentiodynamic polarization curves-from the value of the initial potential E_init_ determined according to the formula E_init_ = E_cop_ − 100 mV. The polarization rate was equal to 1 mV/s and the potential change was in the direction of the anode and once the maximum measuring range reached + 4000 mV, the polarization direction was changed. Based on the study, the following parameters were determined: the corrosion resistance E_corr_ [mV], transpassivation potential E_tr_ [mV], breakdown potential E_b_ [mV], repassivation potential E_cp_ [mV]. The value of polarization resistance Rp and Tafels components b_a_ and b_c_ [mV] were determined with the use of the Tafel method and corrosion current density was calculated using the formula icorr = 0.026/Rp.

All of the electrochemical tests were performed in 250 mL of Ringer’s solution (NaCl—8.6 g/cm^3^, KCl—0.3 g/cm^3^, CaCl_2_ × 2H_2_O—0.33 g/cm^3^) at 37 ± 1 °C and pH 7.4. Subsequently, to simulate inflammation, the pH of the solution was adjusted by controlled injection of a 1M HCl solution until the critical value was pH 5. The electrolyte temperature was kept constant using a Julabo F12-MA cooling/heating circulator (Cole-Parmer, Veron Hills, UK).

The content of aluminum, vanadium and titanium for the M1 samples’ group and additionally Zn for the M2 samples’ group in saline Ringer solutions were determined using inductively coupled plasma atomic emission spectrometer Varian 710-ES. The system is equipped with a OneNeb nebulizer and twister glass spray chamber. The parameters of the analysis were as follows: RF power 1.0 kW, plasma flow 15 L/min, auxiliary flow 1.5 L/min, nebulizer pressure 210 kPa, pump rate 15 rpm, emission lines of Al: λ = 237.312 and 396.152 nm, V: λ = 268.796, 292.401, 309.310 and 311.837 nm, Ti: λ = 334.188, 334.941, and 336.122 nm. The calibration curve method was applied, and the standards were prepared using the matrix (Ringer solution) and single element standard solutions of 1 mg/mL (Merck Millipore, Darmstadt, Germany). Deionized water was obtained using a Millipore Elix 10 system. The results were calculated as an average value of concentrations obtained for all used analytical lines with standard deviation not exceeding 1.7%.

The microscopic observation of the surface of the tested samples after potentiodynamic tests was carried out using a scanning electron microscope (Zeiss, Oberkochen, Germany), equipment with SE (secondary electron) detector. Additionally, the microchemical composition analysis was carried out using energy-dispersive X-ray spectroscopy (EDS) with the point-by-point method.

To determine the surface wettability and contact angles θ analysis measurements, the sessile drop method was performed on both sample groups. The measurements were performed using a test stand, incorporating a Surftens Universal Goniometer (OEG, Frankfurt, Germany) and a PC with Surface 4.5 software. Two measure liquids–distilled water (POCH S.A., Gliwice, Poland) and diiodonomethane (Merck, Warsow, Poland) were used. The wettability angle measurements were carried out using drops of liquid with a volume 1.5 µL at temperature T = 23 ± 1 °C (289 K) placed on the surface of the samples. For both tested sample groups five measurements using both measurement liquids were performed, and the average value was determined. The values of SFE were calculated based on the recorded contact angle. The values of SFE and their components are given in Table 2.

For all physicochemical and electrochemical tests, five samples of botch groups (initial state and with ZnO layer) were undertaken. 

## 3. Results

Based on the microscopic observation (Figure 2a) it ca be concluded, that the ZnO layer deposited by the ALD method was conformal, homogeneous and defect-free (no discontinuities, cracks, pores were detected). The microanalysis of the chemical composition of EDS showed the presence of only zinc Zn and oxygen O.

The columnar growth and columnar orientation of grains were observed, which is typical for ZnO grown by the ALD method [42]. The grains had an elongated shape and were disordered without a dominant direction. The registered XRD pattern had a major diffraction peak of ZnO at an angle (2θ) of 32.1° along (100) plane and other diffraction lines of lower intensity attributed to (002), (101), (102), (103), (112), (2200), (112) and (102) planes of the hexagonal wurtzite ZnO, respectively. Similar results were obtained by Feng at all. [43] and Staszuk at all. [40]. 

On the basis of AFM scans (Figure 3) it can be concluded, that the samples with a ZnO layer (M2) were characterized by higher surface roughness in nanoscale, as compared to the samples in the initial state (M1). 

The mean value of R_a_ parameter for uncoated samples (M1) was R_a_ = 21 nm and for the samples with ZnO layer R_a_ = 62 nm. Additionally, the average distance between the highest peaks analyzed for samples with the ZnO layer assumed similar values.

On the Figure 4 was presented the OCP values changes in time function.

In the neutral Ringer solution of pH = 7.4, the OPC values of samples in the initial state (M1), positively shift direction over all measuring time without noticeable stabilization. The OCP potential of the M1 sample group, tested in Ringer solution of pH = 5 presents almost no change over time. For samples in the initial state M1, the OCP was higher in the solution containing chloride ions (−200 mV_SCE_) compared to that in Ringer solution alone, (−350 mV_SCE_) which is likely to be due to an increased cathodic current. The OCP values close to −350 mV_SCE_ and −200 mV_SCE_ (corresponding to approximately −81 mV_NEH_ and +44 mV_NEH,_, respectively) of the Ti6Al4V alloy for both tests’ conditions, belongs to domains of the passive region of TiO_2_ in the Pourbaix diagram (Figure 5a) indicating the fact that under steady-state conditions, the alloys form a stable oxide layer of TiO_2_ [44,45].

For the samples with the ZnO layer, regardless of the pH of Ringer’s solution, the OCP potential gradually increased. It was associated with a thickening of the oxide film and/or corrosion products layer on the samples. Additionally, the OCP values of that same pH of Ringer solution were higher for the samples with the ZnO layer as compared to the samples in the initial state. In the Ringer solution of pH = 7.4, the OCP potential of the M2 sample group presents some oscillations. This can be related to some instability of ZnO oxide films in test conditions.

The mean value of OCP was close to −109 mV_SCE_ (+135 mV_NEH_), which is more noble, compared to the M2 samples, tested in Ringer solution with the addition of chloride ions, where the mean value was −223 mV_SCE_ (+21 mV_NEH_). According to the Pourbaix diagram (Figure 5b), zinc is present as hydrated Zn^2+^(aq), over both ranges of the pH values of Ringer’s solution, and the OCP values belong to biological standard reduction potentials (~820 mV to ~−670 mV) [46,47]. The area of zinc passivation (the field of solid corrosion products) occurs for a narrow range of pH values from 9 to 11. However, the domains of the passive zinc region can be expanded to pH 6–11 by the adsorption of carbon to the zinc surface [48].

The results of the potentiodynamic test in form of anodic polarization curves, which can be used to confirm the interpretation of OCP behavior, are presented in Figure 6 and the pitting corrosion parameters are given in Table 3.

For M1 for both variants of the Ringer’s solution pH, higher values of the b_c_ (b_c_ = 67 ± 7, pH—7.4 and b_c_ = 69 ± 8, pH = 5) were obtained compared to b_a_ (b_a_ = 54 ± 6, pH—7.4 and b_a_ = 69 ± 8, pH = 5). A reverse tendency has been reported for the samples with the ZnO layer. Higher values of anodic reaction (b_a_ = 256 ± 14 mV, pH—7.4 and ba = 207 ± 19 mV, pH = 5) as compared to the cathodic reaction were recorded (b_c_ = 88 ± 9 mV, pH—7.4 and pH = 5). On the basis of the recorded curves it was established that the corrosion resistance was different depending on the surface condition and Ringer’s solution pH. Only for the samples in initial state M1, tested in neutral Ringer solution of pH = 7.4, the was the existence of transpassivation potential E_tr_ measured. The mean value was approximately E_tr_ = 2410 ± 120 mV. When the pH of the corrosion solution was decreased to 5.0, the hysteresis loops were recorded and the existence of the breakthrough potential E_b_ and the repassivation potential E_cp_ was recorded, of which the mean values were E_b_ = 1800 ± 89 mV and E_cp_ = 827 ± 74, respectively. For both test conditions, the voltammetric curve of samples in the initial state (M1) shows an active to passive transition with current densities of about −350 µA/cm^2^ and a similar plateau from ~+10 to + 2000 mV followed by a steady increase. A similar character of the progress of anodic polarization curves was recorded for the samples after surface modification. However, depositing the ZnO layer (M2) by the ALD method, regardless of the solution pH, a positive increase in the values of breakdown potential compared to values of transpassivation potential of samples in initial state (M1) was found, which favorably affects the corrosion resistance. In the case of samples with the ZnO layer, recorded curves were characterized by an active to passive transition with current densities of about −400 µA/cm^2^ without an explicit area of plateau. For the M2 sample group, the chloride ions have a negative influence on the electrochemical parameters due to their higher corrosivity. The mean values of breakdown potentials were E_b_ = 2958 ± 78 mV (pH = 7.4) and E_b_ = 2600 ± 102 mV (pH = 5). Additionally, with the deposited ZnO layer, the polarization resistance also increased from 469 ± 87 kΩ∙cm^2^ (pH = 7.4) and 568 ± 111 kΩ∙cm^2^ (pH = 5) for the uncovered samples to a value of 10973 ± 212 kΩ∙cm^2^ (pH = 7.4) and 1563 ± 87 kΩ∙cm^2^ (pH = 5) for samples with ZnO layer. A similar tendency for 316 stainless steel was obtained by Basiaga et al. [49].

EIS studies were performed to obtain an understanding of the influence of the surface modification and electrochemical test conditions on the corrosion mechanism of the Ti6Al4V titanium alloy. Figure 7 shows the EIS spectra recorded for both sample groups in the form of Nyquist and Bode plots.

The Nyquist diagram for both sample groups, tested in Ringer’s solution alone and with chloride ions, presented fragments of semi-circles, which is a typical response of a thin layer (Figure 7a,c,e,g). The presented semi-circles were deformed to a different degree, which were obtained also in previous tests [50]. The samples with the ZnO layer (M2), regardless of the pH value of Ringer solution, were characterized by the biggest angle of inclination of the curves to the ordinate axis, which pointed to better corrosion behavior compared to uncoated samples (M1). Increasing radius of semi-circles, presented on Nyquist plots, pointed to increased impedance values with the ZnO layer deposition. The lowest value of the maximum value of phase displacement at a broad range of frequencies, which are presented in the Bode diagram, was obtained for samples in the initial state (M1), tested in a neutral Ringer solution and the mean value was approximately Θ = 45°. For other samples the values were similar and were in the range of Θ = 70–80°. The lower of the wide range of high phase shift angles should be associated with instability of their oxide films, which correspond to the analysis of OCP diagrams (Figure 4). For all sample variants, the inclinations log|Z| at the whole scope of frequency changes is close to −1, which indicates the capacity character of the porous layer.

For the samples in initial state (M1), tested in neutral Ringer solution pH = 7.4 and with chloride ions, pH = 5, an equivalent circuit with two time constants was used to analyse the EIS data, which indicates the occurrence of two sublayers. A similar equivalent circuit was noted for samples with a ZnO layer (M2), tested in a solution of pH = 5. The equivalent circuit (Figure 8a,c) consists of C_pore_/CPE_pore_ (capacity of the double layer-porous surface) and R_pore_ (resistance of double layer-porous surface), which are representatives of the electrical porous layer, whereas CPE_dl_ and R_ct_, represent the resistive and non-ideal capacitive behavior of the passive film.

According to this model, the two R, CPE/C elements are connected in series with the Ringer solution resistance R_s_. The model describes the formation of two loops in a Nyquist plot. The activity of the oxide layer/deposited ZnO layer is characterized by a high-frequency loop, which diameter depends on the potential. This is determined by the serial resistance R_pore_ (electrolyte resistance in the porous phase) and the space charge capacitance C_pore_/CPE_pore_ (capacity of the double layer-porous surface). The low-frequency loop corresponds to the surface boundary of the oxide layer solution, and the R_ct_ (characterizes the electric charge transfer resistance at the boundary of phases) and CPE_dl_ is used to describe the low-frequency region (11–0.001 Hz) [51]. The mathematical impedance model of the above system is also presented in the Equation (1) for M1 samples group and in Equation (3) for M2 samples group (pH = 5). Impedance spectra (Figure 7) obtained for the samples with the ZnO layer, tested in neutral Ringer solution pH = 7.4, were interpreted by comparison to the model, where the passive film is considered to have a porous structure and to show non-ideal capacitive behavior (Figure 8b). R_ad_ is the charge transfer resistance of the electrochemical process taking place inside the pore, and C_ad_/CPE_ad_ are the capacity of the adsorption layer. The mathematical impedance model of the above system is also presented in Equation (2).
(1)$$\;Z=R_{s}+\frac{1}{\frac{1}{R_{p}}+\;Y_{01}{\left(j\omega \right)}^{n_{1}}}+\;\frac{1}{\frac{1}{R_{\mathrm{ct}}}+\;Y_{02}{\left(j\right)}^{n_{2}}}\;$$
(2)$$Z=R_{s}+\frac{1}{\frac{1}{R_{\mathrm{pore}}}+\;Y_{1}{\left(j\right)}^{n_{1}}}+\;\frac{1}{\frac{1}{R_{ct}}+\;Y_{2}{\left(j\right)}^{n_{2}}}$$
(3)$$Z=R_{s}+\frac{1}{\frac{1}{R_{\mathrm{ad}}}+\;jC_{ad}}+\frac{1}{\frac{1}{R_{\mathrm{pore}}}+\;jC_{pore}}+\;\frac{1}{\frac{1}{R_{ct}}+\;Y_{2}{\left(j\right)}^{n_{2}}}$$

The numerical values of the electrical model of equivalent circuit for the investigated materials are summarized in Table 4.

The highest value of R_ct_ resistance was obtained by samples with the ZnO layer, which confirmed to that the applied protective layers on the material contributed to the improvement of anticorrosive properties. The values obtained for the M1 sample group, for both test conditions were similar and were in the range from 393 kΩ cm^2^ (pH = 7.4) to 449 kΩ cm^2^ (pH = 5). Based on this, it can be concluded that the ZnO layer was characterized by good quality and compact structure.

In order to obtain more information about corrosion characteristics of both tested sample groups, after the potentiodynamic and electrochemical tests the samples were subjected to microscopic observation using the scanning electron microscope. Examples of results for samples with the ZnO layer are presented in Figure 9.

For all experimental groups, the amorphous sponge-like corrosion products cover most of the surface (Figure 9a). Additionally, on the surface of the samples, corrosion products are also visible in the form of a small amount of laminar crystals (Figure 9c and Figure 10).

The EDS results show that the primary elements of the sponge-like products were Zn and O (Figure 9b), while the primary elements of laminar crystals had the same chemical species but with addition of Cl ions (Figure 9d).

According to the results obtained by inductively coupled plasma atomic emission spectrometry (Table 5), it can be concluded that the ZnO layer is a good protection against the release of Ti6Al4V alloy element ions into the corrosive environment (human organism).

The analysis of the amount of ions released into the neutral Ringer’s solution for M1 sample groups, indicates a trace presence of elements. However, reducing the pH of the solution to the critical measurement increases the amount of titanium ions released into the solution. The deposition of the ZnO oxide layer by the ALD method inhibits the release of Ti6Al4V alloy elements tested both in Ringer’s solution and at pH = 7.4 and pH = 5, while releasing zinc ions. The process is intensified by reducing the pH of the corrosion solution.

The result of the measurements of the surface wettability and calculation of surface free energy are presented in Table 6 and the examples of drops placed on the surface of both sample groups and the scheme of the change of the contact angle in time function are presented in Figure 11.

Based on the results obtained, it was found that there is an influence of the zinc oxide layer ZnO deposition on the surface of the Ti6Al4V alloy for the wetting angle compared to the samples in the initial state. There were significant differences in the values of the wetting angle for M1 and M2 sample groups. For the samples in the initial state M1, the mean value of the water wetting angle was approximately θ = 63 ± 3°, which pointed to hydrophilic character of the surface. The surface modification by the ALD method affected the increase in wetting angle. For the samples with the ZnO layer M2 the angle value was θ = 110 ± 2°. The value of contact angle of more than 90° indicates the hydrophobic character of the surface. The scheme of contact angle changes in time function, showed an insignificant decrease in the value of the contact angle during the t = 60 s measurement. Additionally, the obtained values of contact angle, using diiodonomethane as a measure liquid, were similar for both sample groups and were in the range from θ = 45 ± 2° for M1 and θ = 54 ± 2° for M2, respectively. Furthermore, the values of surface free energy were similar to γ_S_ = 43–42 mJ/m^2^. For both sample groups, higher values of apolar components of SFE were observed, as compared to the polar components. Based on this it can be concluded that all tested samples exhibited a greater affinity to apolar groups of SFE than to polar ones. However, in case of the samples with the ZnO layer (M2) the differences in the values between polar and apolar components were bigger.

## 4. Discussion

First, the surface quality of the ZnO layer deposited by the ALD method was examined. On the basis of the results obtained it can be concluded that a chemically homogeneous defects-free layer was obtained. Additionally, it was found, that the modification led to the development of nanoscale surface of Ti6Al4V alloy (Figure 3).

Different corrosion resistance values were obtained for the uncoated samples (M1) and samples with the ZnO layer (M2). Additionally, it was found that the pH of Ringer’s solution had a significant influence on the corrosion parameters. Generally, it is known that the electrode potential in a metal-electrolyte system reflects the state of the surface of the metal that is immersed in the corrosive medium. Based on the results obtained from the open-circuit potential OCP measurements (Figure 4), it was recorded that for the samples in initial state (M1), tested in Ringer’s solution with the addition of chloride ions (pH = 5) the more noble values were obtained, compared to the samples tested in the neutral Ringer solution (pH = 7.4). The reverse tendency has been reported for the samples with the ZnO layer. Additionally, for the samples with the ZnO layer, regardless of test conditions, more positive values of OCP were recorded. According to the results obtained by Dai et al. [51] and Bhola et al. [44], it can be concluded that, the OCP shift in the more noble direction suggests the formation of a passive layer, which is a barrier for titanium alloys dissolution and reduces the corrosion rate. The shift for samples with the ZnO layer (M2) can be associated with thickening of the oxide film and/or corrosion product layer on the samples surface. The open-circuit potential increase shows that the tested samples become thermodynamically more stable with time.

The results of the potentiodynamic test confirm the interpretation of OCP behavior. The order of the OCP (Figure 6) is consistent with that in Figure 4, and a similar tendency was obtained by Zhang et al. [52]. More positive values of OCP were recorded (for both test conditions) for the samples with the ZnO layer (M2), which led to a higher potential associated with a higher anodic reaction rate. This is confirmed by the values obtained using the Tafel method (Table 3). Based on the results obtained by Blackwood et al. [53], it can be concluded that decreasing corrosion rate in effect of shift values of OCP to positive regions is associated with reducing the driving force of the cathodic reaction and increasing the thickness of the passive layers. Based on the results obtained from potentiodynamic tests, for samples after surface modification a favorable increase in corrosion resistance was observed (Table 4, Figure 6). The samples with ZnO Layer were characterized by higher values of corrosion potential and transpassivation/breakdown potentials and polarization resistance. According to the results obtained by the EIS method, it can be conceded that the deposition of the ZnO layer on the Ti6Al4V led to high electrochemical stability of the oxide layer as evidenced by the relatively high value of the charge transfer resistance Rct. Corrosion of zinc involves the anodic dissolution of zinc, according to Equation (4):(4)$$Zn\;\to \;Zn^{2+}{\left(aq\right)}^{+}+e$$

For the M2 sample group, the microscopic investigations revealed the presence of the corrosion products in form of the amorphous sponge-like products (Figure 9a) and a certain amount of laminar crystals (Figure 9c). According to the results obtained by Yin et al. [54], it can be concluded that the sponge-like products are mainly comprised of Zn_5_(CO_3_)_2_(OH)_6_ and the laminar crystals should be consisting of Zn_5_(OH)_8_Cl_2_·H_2_O (Figure 11). The OCP potential of M2 samples group, tested in neutral Ringer solution presents some oscillations. The shape of the OCP curve can be related to some instability of ZnO layer. The instability of the oxide layer may be caused by its active dissolution and the simultaneous processes of its reconstruction. This proves only partial dissolution of zinc oxide in Ringer’s solution and deposition of corrosion products on the surface of the tested sample, which constitutes anti-corrosion protection, which confirms the highest value of OCP in the studied group.

Additionally, the obtained forms of the characteristics of EIS spectra indicate the presence of a porous oxide layer. In the case of these samples, lower values of the maximum value of phase displacement at a broad range of frequencies Θ = 45° were recorded, which also pointed to instability of their oxide films, which corresponds to the analysis of OCP diagrams (Figure 5). Based on the results obtained by Basiaga et al. [55] it can be concluded that this is due to the reaction at the layer–solution interface.

Additionally, Thomas et al. [56] showed that the reduction in the rate of zinc corrosion is caused by a reduced rate of the cathode reaction. Additionally, surface oxides are an effective anti-corrosion barrier. On this basis, it can be concluded that metallic zinc in physiological environments with a pH of 7.4 will dissolve over time, which may explain the presence of numerous corrosion products on the surface of M2 samples. Results of ICP–AES analysis pointed to increase number of Zn ions in Ringer solution after corrosion tests, which was intensified by reducing the pH of the corrosive solution. The Zn^2+^ released from the ZnO layer to the surrounding extracellular space may be biointegrated with human tissue or encapsulated in a dense fibrous tissue that may lead to inflammatory. The cellular response to the Zn^2+^ have significant impart the healing process and biocompatibility of zinc oxide layer [57]. However, results obtained by atomic emission spectrometry showed that the ZnO layer is a good protection against the release of Ti6Al4V alloy element ions into the corrosive environment (human organism). For the uncoated samples (M1), tested in Ringer solution of pH = 5, an increased number of released Ti ions can be observed.

The surface modification by ZnO layer deposition by ALD methods leads to the improvement of the corrosion resistance properties of Ti6Al4V titanium alloy. In the literature there are some reports about that a satisfactory corrosion resistance that has been also obtained as a result of surface modification using PVD (Physical Vapour Deposition) [58], CVD (Chemical Vapour Deposition) [59], anodic polarization [60] and many other methods [61,62]. However, an important advantage of the ALD method over the others is the possibility of covering very complex implants shape.

The contact angle measurements showed that, the surface modification by the ALD method effected in the increase in wetting angle. Similar trends were noted by Basiaga et. al.for the ZnO layer [55,62] and a previous study about ZrO_2_ oxide layers deposited by ALD methods [24]. It was concluded from the comparative studies that the surface wettability conversion reactions on the tested samples are directly related to the morphology of the samples, especially to surface roughness. The AFM surface roughness calculation showed that the samples with the ZnO layer M2 were characterized by close to three-fold higher values of surface roughness parameters in nanoscale (Figure 3) as compared to the samples in the initial state M1. Based on this it was found that deposition of the ZnO layer by the ALD method leads to the increase of specific surface in nanoscale. Presented wetting states can be categorized in the Wenzel state [29,63]. According to the Wenzel condition, the reduction of the surface energy of the drop (due to surface tension forces) is associated with the formation of a spherical shape. According to Young’s Equation (5) ($\gamma_{sg}$, $\gamma_{sl}$, $\gamma_{lg}$ are interfacial tensions of the solid–gas, solid–liquid and liquid–gas interference, respectively, Θ is the contact angle and r is the roughness ratio), the wettability depends on roughness effects. Similar results were obtained by Wang et al. [63] for Al_2_O_3_ thin films. The results reveal that the ALD method can be used to effectively modify the surface hydrophilicity/hydrophobicity of the ZnO layer.
(5)$$\cos\Theta =r\frac{\gamma_{sg}-\gamma_{sl}}{\gamma_{lg}}=r\cos\Theta$$

However, Wang [64] showed that the hydrophobic properties of Al_2_O_3_ films are attributed to carbon residue. The literature data suggested that metal oxides are generally intrinsically hydrophilic due the presence of metal cations, oxygen anions and hydroxyl group on the surface [64]. A similar conclusion was obtained by Bae et al. [65] showing that the hydrophobic nature of oxide layers deposited by the ALD method may be intrinsic to the result of the presence of adsorbed hydrocarbons. Increased hydrophobicity upon atmospheric exposure was observed for ZrO_2_, Al_2_O_3_, TiO_2_, SiO_2_ and CeO_2_ metal oxide surfaces. In future studies, an XPS analysis should be performed to confirm the literature reports. The hydrophobic character of the surface was a favorable phenomenon because lower wetting leads to reduced protein adsorption and limits the formation of microorganisms on the implant surface and in effect can provide better protection against biocorrosion.

## 5. Conclusions

The sum up, the pH of Ringer’s solution has a significant effect on the corrosion resistance of uncoated M1 samples and samples with the ZnO layer. For the samples tested in Ringer’s solution of pH = 5, lower corrosion resistance and an increased number of ions released to the medium were recorded. For samples with the ZnO layer, increased wetting angle and corrosion resistance, including pitting corrosion resistance and higher barrier properties, were recorded. The hydrophobic character of the surface and high value of surface free energy are associated with better corrosion resistance and can lead to reduced bacterial adhesion thereby, in effect, reducing biocorrosion. Additionally, along with the deposition of the ZnO layer, the amount of titanium ions released to the corrosion solution was reduced. The ZnO layer deposited on the Ti6Al4V substrate had better physicochemical properties compared to the uncoated samples. In the future, in vitro biological studies should be carried out to confirm the relationship between the chemical character of the surface treatment as well as cellular response and bacterial adhesion.

## Figures and Tables

**Figure 1 materials-14-00230-f001:**
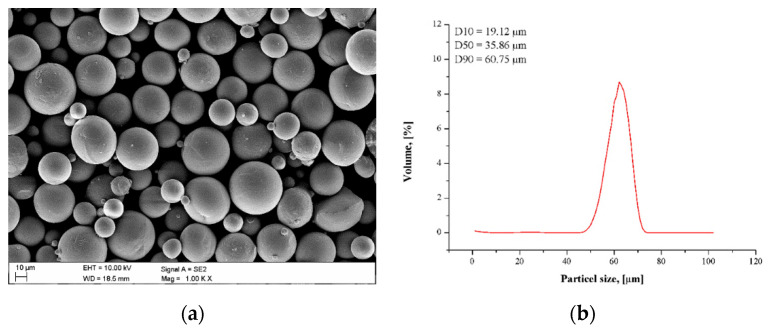
Ti6Al4V titanium alloy powder characterization (**a**) scanning electron microscope (SEM) image of powder morphology, (**b**) particle size distributions.

**Figure 2 materials-14-00230-f002:**
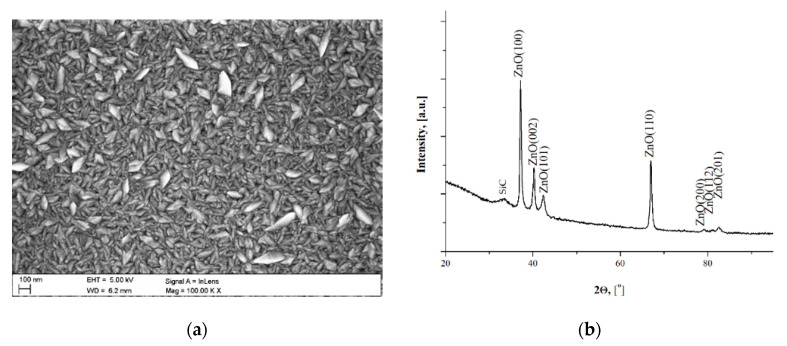
Oxide layer ZnO characterization (**a**) surface morphology scanning electron microscopy (SEM); (**b**) X-ray energy dispersive plot.

**Figure 3 materials-14-00230-f003:**
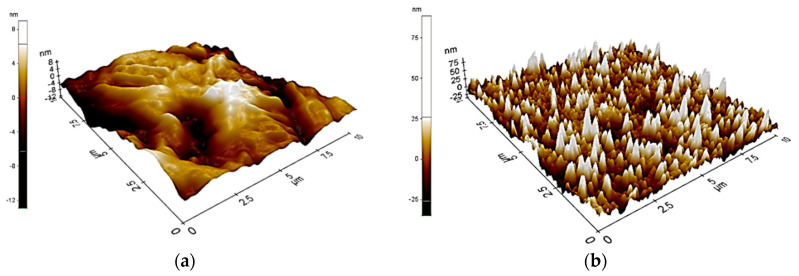
Examples surface topography analysis (**a**) 3D view of atomic force microscopy (AFM) scan from 10 × 10 µm^2^ M1; (**b**) 3D view of AFM scan from 10 × 10 µm M2.

**Figure 4 materials-14-00230-f004:**
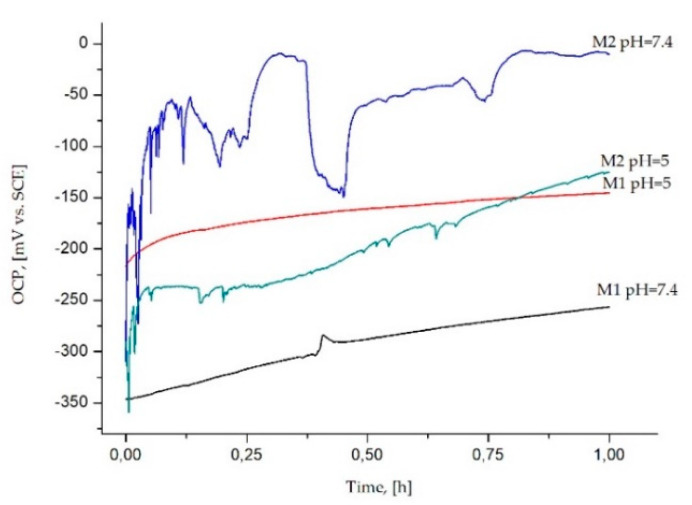
Open circuit potential for M1 and M2 sample groups in Ringer solution of pH = 7.4 and pH = 5, exposure for t = 1 h.

**Figure 5 materials-14-00230-f005:**
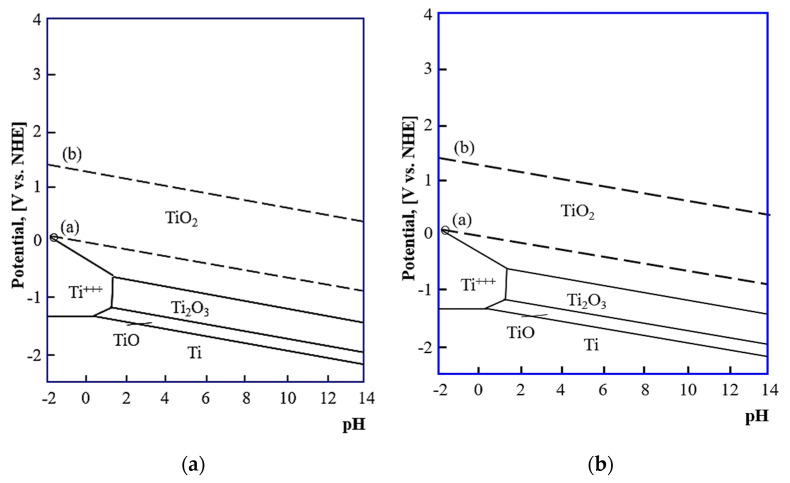
Pourbaix diagram for (**a**) titanium at T = 37 °C; (**b**) for zinc.

**Figure 6 materials-14-00230-f006:**
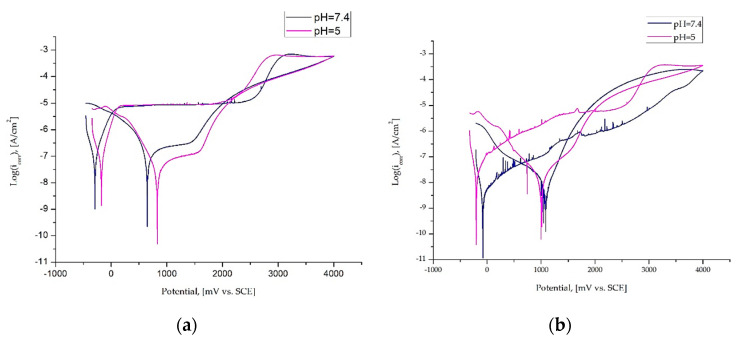
Example of polarization curves (**a**) M1 samples group; (**b**) M2 sample group.

**Figure 7 materials-14-00230-f007:**
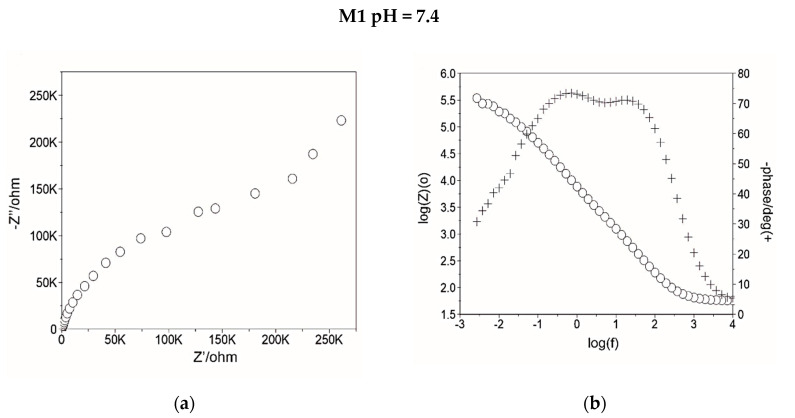
Examples of results of electrochemical impedance spectroscopy (EIS) test for M1 and M2 samples (**a**,**c**,**e**,**g**) Nyquist diagram; (**b**,**d**,**f**,**h**) Bode diagram.

**Figure 8 materials-14-00230-f008:**
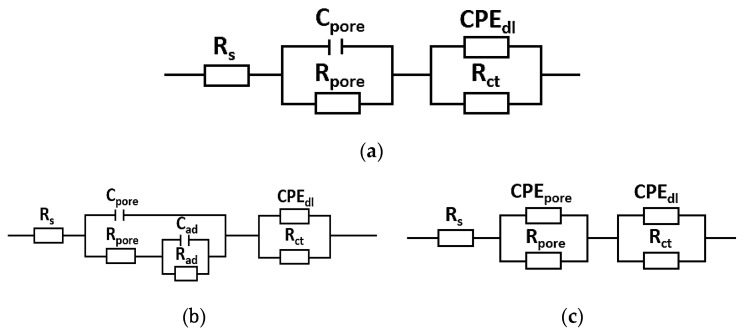
Electric substitute scheme (**a**) M1 pH = 7.4 and pH = 5; (**b**) M2 pH = 7.4; (**c**) M2 pH = 5.

**Figure 9 materials-14-00230-f009:**
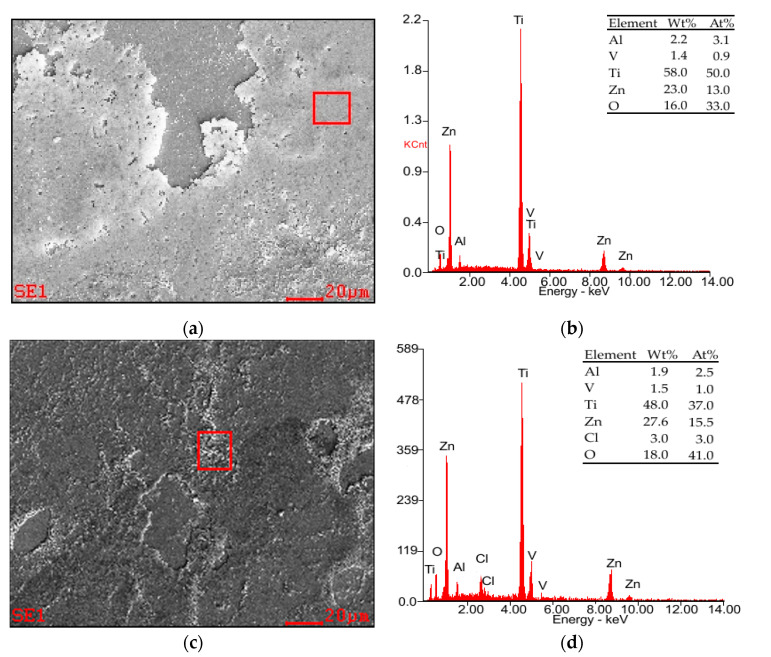
Examples of SEM observations of the M2 sample group (**a**) sponge-like corrosion products; (**b**,**c**) corrosion products in form of laminar crystals (**d**) energy-dispersive X-ray spectroscopy (EDS) results for marked area in Figure 9c.

**Figure 10 materials-14-00230-f010:**
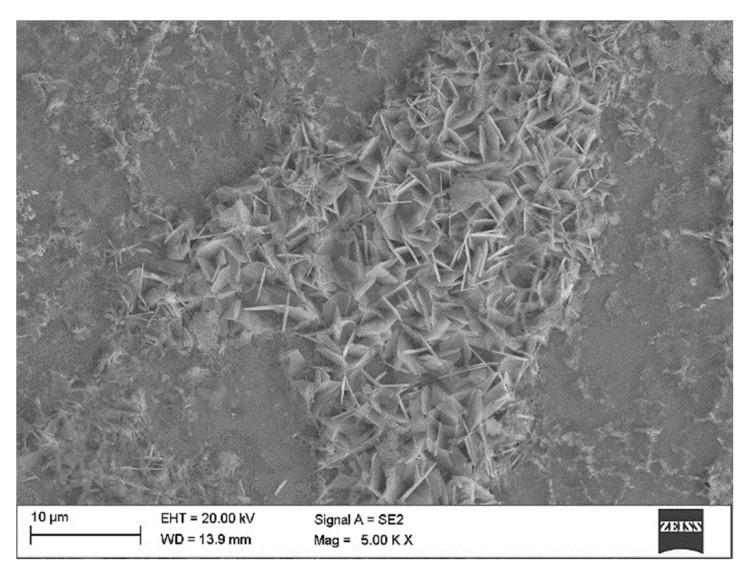
Examples of SEM observations of the M2 sample group after corrosion tests.

**Figure 11 materials-14-00230-f011:**
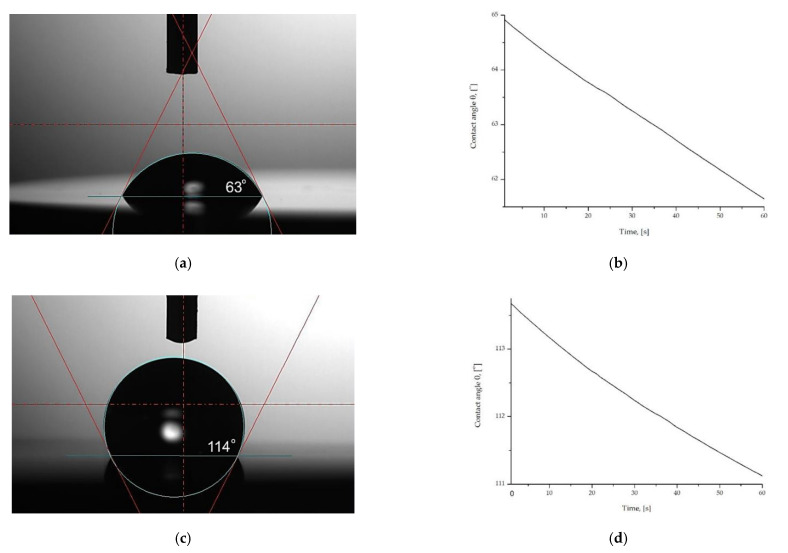
Results of contact angle measurements (**a**) drop M1; (**b**) contact angle values changes in time function M1; (**c**) drop M2; (**d**) contact angle values changes in time function M2.

**Table 1 materials-14-00230-t001:** Chemical composition of the Ti6Al4V titanium alloy.

Element, wt. (%)	Al	V	Fe	O	C	Ni	H	Ti
Declared	5.50–6.50	3.50–4.50	≤0.25	≤0.13	≤0.08	≤0.05	≤0.012	rest
ISO 5832-3	5.5–6.75	3.5–4.5	≤0.3	≤0.2	≤0.08	≤0.05	≤0.015	rest
Powder	5.60	3.70	≤0.1	≤0.2	-	-	-	rest
As-Fabricated	5.90	3.95	≤0.1	-	-	-	-	rest

**Table 2 materials-14-00230-t002:** The values of surface free energy (SFE) and their components for distilled water and diiodonomethane used in the Owens–Wendt method [24].

Measure Liquid	SFE, (mJ/m^2^)		
	γ_L_	$\gamma_{L}^{d}$	$\gamma_{L}^{P}$
Distilled water	72.80	21.80	51.00
Diiodonomethane	50.80	50.80	0

**Table 3 materials-14-00230-t003:** Results of pitting corrosion test—mean values and standard deviations. E_corr_—corrosion potential, E_tr_—traspassivation potential, E_b_—breakdown potential, E_cp_—repassivation potential, b_a,_ b_c_—Tafels componets, R_p_—polarization resistance, i_corr_—current density.

pH	Name	E_corr_, (mV)	E_tr_, (mV)	E_b_, (mV)	E_cp_, (mV)	b_a_ (mV)	b_c_, (mV)	R_p_, (kΩ ·cm^2^)	i_corr_, (µA/cm^2^)
7.4	M1	−298 ± 21	2410 ± 120	-	-	64 ± 9	67 ± 7	469 ± 87	0.028 ± 0.002
5		−175 ± 11	-	1800 ± 89	827 ± 74	54 ± 6	69 ± 8	568± 111	0.026 ± 0.005
7.4	M2	−85 ± 12	-	2958 ± 78	1120 ± 98	256 ± 14	88 ± 9	10973 ± 212	0.003 ± 0.001
5		−205 ± 14	-	2600 ± 102	1000 ± 124	207 ± 19	88 ± 14	1563 ± 120	0.017 ± 0.001

**Table 4 materials-14-00230-t004:** Results of EIS test.

Name	pH	E_opcr_, [mV]	R_s_, [Ωcm^2^]	R_ad_, [kΩcm^2^]	C_ad_, [µF]	CPE_ad_		R_pore_, [Ωcm^2^]	C_pore,_ [µF]	CPE_pore_, [mV]		R_ct_, [kΩcm^2^]	CPE_dl_, [mV]	
						Y_0_, [kΩcm^−m^ s^−n^]	n_0_			Y_1_, [kΩcm^−m^ s^−n^]	n_1_		Y, [Ωcm^−m^ s^−n^]	n_2_
M1	7.4	−320	85	-	-	-	-	38	75	-	-	449	0.4178 × 10^−4^	0.78
M1	5	−271	75	-	-	-	-	14	45	-	-	393	04851 × 10^−4^	0.72
M2	7.4	−133	86	40	2	-	-	81	13	-	-	885	0.6590 × 10^−5^	0.78
M2	5	−128	74	-	-	-	-	204		0.3784 × 10^−5^	0.90	1892	0.1573 × 10^−4^	0.84

**Table 5 materials-14-00230-t005:** Results of inductively coupled plasma atomic emission spectrometry (ICP–AES) analysis.

No.	pH	Name	Release Ions, (mg/L)			
			Al	V	Ti	Zn
1	7.4	M1	<0.01	<0.01	<0.01	-
2	5	M1	0.02	<0.01	0.14	-
3	7.4	M2	-	<0.01	<0.01	0.15
4	5	M2	-	<0.01	<0.01	0.44

**Table 6 materials-14-00230-t006:** Results of contact angle measurements and calculated SFE.

No.	Name	Wetting Angle, (°)		Surface Free Energy, (mJ/m^2^)		
		Distilled Water	Diiodonomethane	γ_S_	$\gamma_{d}^{S}$	$\gamma_{p}^{S}$
1	M1	63 ± 3	45 ± 2	42	23	19
2	M2	110 ± 2	54 ± 1	43	42	1

## Data Availability

Not applicable.

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
