# Peer review of "The Influence of ZnO Oxide Layer on the Physicochemical Behavior of Ti6Al4V Titanium Alloy"

_materials, 2021, doi:10.3390/ma14010230_

Round 1

Reviewer 1 Report

Manuscript:

The influence of ZnO oxide layer on the physicochemical behavior of the Ti6Al4V titanium alloy

Comments

The topic is interesting. Over the last decade, zinc oxide has been used as a coating material for orthopedic and dental implants, see for example J Biomed Mater Res A. 2015 Mar;103(3):981-9.doi: 10.1002/jbm.a.35241. Epub 2014 Jun 4
Furthermore there are review papers on the topic: “New Ti-Alloys and Surface Modifications to Improve the Mechanical Properties and the Biological Response to Orthopedic and Dental Implants: A Review”, BioMed Research International, 2016 |Article ID 2908570 | https://doi.org/10.1155/2016/2908570.

The authors of this work propose a modification of the Ti6Al4V alloy surface by deposition of a ZnO layer by Atomic Layer Deposition (ALD) method. They show that the ALD method allows deposition of extremely conformal and high-quality barrier layers with controllable thickness, even on complex three dimensional surfaces. The main comments would be that the authors fail to compare their method to other methods in their discussion.

Minor comments:

  1. Introduction, line 62, “One of the most popular metal oxide layers are: silver Ag [19], copper oxide 63 CuO [20], gold Au [21], titanium oxide TiO2 [22], aluminum oxide Al2O3 [23], zirconia oxide ZrO2 [24].” Ag and Au are not metal oxides, but Auâ‚‚O₃ and Ag2O are metal oxides.
  2. Line 67 define MSCs.
  3. Line 141 The thickness of the layer was approximately ~ 200 nm. 200 +-?nm?
  4. Quality of figures: Fig2(b), 4, 6, 11 (b) and (d) should be improved.
  5. In places et al is written et all
  6. A number of typing errors should be corrected.

Author Response

26.12.2020

Anna Woźniak

Silesian University of Technology

Konarskiego 18A st. 44-100 Gliwice

Dear Reviewer,

Serdecznie dziÄ™kujemy za zmiany w naszym artykule pt. „WpÅ‚yw warstwy tlenku ZnO na wÅ‚aÅ›ciwoÅ›ci fizykochemiczne stopu tytanu Ti6Al4V”. Doceniamy pracÄ™, którÄ… wykonaÅ‚eÅ›, aby ulepszyć nasz artykuÅ‚. WprowadziliÅ›my niezbÄ™dne poprawki, biorÄ…c pod uwagÄ™ komentarze. pragniemy wysÅ‚ać poprawionÄ… wersjÄ™ i odpowiedzieć na postawione zarzuty.

- GÅ‚ównym komentarzem byÅ‚oby to, że autorzy w dyskusji nie porównujÄ… swojej metody z innymi metodami.

Informacje o odpornoÅ›ci korozyjnej modyfikowanego różnymi metodami stopu tytanu Ti6Al4V zamieszczono w artykule - rozdziaÅ‚ Dyskusja, wiersz 457.

-Line 141 Grubość warstwy wynosiła około ~ 200 nm. 200 + -? Nm?

Grubość warstwy ZnO osadzonej metodÄ… ALD wynosiÅ‚a okoÅ‚o 200 ± 20 nm. UwzglÄ™dniono informacje w rozdziale „MateriaÅ‚y i metody”. Wiersz 141.

Uwzględniono wszystkie sugestie dotyczące poprawy jakości rycin.

Mamy nadzieję, że poprawki spełniły Twoje oczekiwania. Życzymy dużo zdrowia i cierpliwości w tych trudnych czasach.

Z poważaniem,

Anna Woźniak

Reviewer 2 Report

The paper is well written and I recommend the publication after the minor revisions.

  1. page 4 line 147: the surface morphology and topography were examined .....".There are some others errors. 
  2. Fig.3 - scan was performed on 10 x 10 um2; no um!
  3. Please add some explanation about the fluctuations of the OCP of M2 in Ringer of pH=7.4 - Fig.4. 7.4 is a neutral pH and it is strange to see these fluctuations. It is not enough to write "....instability of their oxide...".    
  4. No elemental composition can be found.
  5. In fig. 5, it is a peak around of 35 deg - what is it here? 
  6. Please add the card no used for XRD identification (JCPDS or ICCD).
  7. page 6 line 236: please correct Ra using subscript.     
  8. Please add in legends of table 3 the definition of each electrochemical parameter.  
  9. How many replicates were used for the corrosion tests and others tests? 

Author Response

26.12.2020

MSc Anna Woźniak

Silesian University of Technology

Konarskiego 18A st. 44-100 Gliwice

Dear Reviewer,

Thank you very much for the amendments to our article entitled “The influence of ZnO oxide layer on the physicochemical behavior of the Ti6Al4V titanium alloy”. We appreciate the work you've done to make our article better. We've made the necessary adjustments, taking into account the comments. we wish to send out a revised version and respond to the allegations made:

  • Please add some explanation about the fluctuations of the OCP of M2 in Ringer of pH=7.4 - Fig.4. 7.4 is a neutral pH and it is strange to see these fluctuations. It is not enough to write "....instability of their oxide...".    

The OCP potential of samples with ZnO layer, tested in neutral Ringer solution presents some oscillations. The shape of OCP curve can be related to some instability of ZnO layer. The instability of the oxide layer may be caused by its active dissolution and the simultaneous processes of its reconstruction. It proves only partial dissolution of zinc oxide in Ringer's solution and deposition of corrosion products on the surface of the tested sample, which constitutes anti-corrosion protection, which confirms the highest value of OCP in the studied group. The respond was include in article (section Discussion; Line 433).

  • No elemental composition can be found.

The elemental compositions of powder material and final elements were given in Table 1 (Line 108). The information of chemical composition of ZnO layer was presented in Line 219. The elemental compositions observed corrosion products were presented in table on Figure 9 – line 356.

  • In fig. 5, it is a peak around of 35 deg - what is it here? Please add the card no used for XRD identification (JCPDS or ICCD).

The phase identification of ZnO layer were detected using ICCD card number 01-073-8765 (in Attachment) . The peak around 35 deg. Is associated with SiC substrate material (according to the JCPDS Card No. 73-1665) – Baig U.; Gondal M.A.; Dastageer M.A.; Khalil A.B.; Zubair S.M.; Photo-catalytic deactivation of hazardous sulfate reducing bacteria using palladium nanoparticles decorated silicon carbide: A comparative study with pure silicon carbide nanoparticles, J of photochemistry and photobiol. B, Biology. 2018, 187, doi:10.1016/j.photobiol.2018.08.010.

XRD analysis for the ZnO layer was performed using a Co lamp. In order to demolish the disturbances from the Ti6Al4V substrate material, the layer was applied to the SiC substrate. A Cu lamp is preferred for XRD Ti6Al4V analysis. The information’s were presented in the article – line 154.

  • How many replicates were used for the corrosion tests and others tests? 

For all physicochemical and electrochemical test five samples of botch groups (initial state and with ZnO layer) were subjected. The information was included in article (section Materials and Methods, line 213).

Additionally, all suggestions for improving the text were included.

We hope that the adjustments made to meet your expectations. We wish you a lot of health and patience in these difficult times.

Sincerely

Anna Woźniak
